🔓 | **Open Peer Review** | Genetics and Molecular Biology | Methods and Protocols

# An effective strategy for identifying autogenous regulation of transcription factors in filamentous fungi

**Longguang Qin,[1] Shuhui Guo,[1] Ang Li,[1] Lu Fan,[1] Kaeling Tan,[1,2] Koon Ho Wong[1,3,4]**

**ABSTRACT** Transcription factors (TFs) play a crucial role in regulating gene expression in living organisms, and any malfunction in their functions can lead to severe physiological consequences. The regulation of TFs occurs at multiple levels, with autogenous regulation being a common mode of regulation where the expression of a TF gene is dependent on its own function. Although bioinformatics analysis of promoter sequences is frequently used to predict autogenous regulation events, it is not entirely reliable. In this study, we present a simple yet effective strategy to identify the autogenous regulation of a TF. We have demonstrated the effectiveness of this method on three fungal TFs (the Carbon Catabolite Repressor CreA and two uncharacterized TFs of secondary metabolism) and have generated a set of plasmids to facilitate the construction of experimental strains in fungi. Our method can be applied to both positive and negative-acting TFs and generalized for other organisms. Hence, this work provides a reliable and straightforward method for identifying autogenous regulation events, which is useful for understanding TFs' functions.

**IMPORTANCE** Transcription factors (TFs) play a crucial role in deciphering biological information from the DNA of living organisms. Improper regulation of their functions can disrupt cellular physiology and lead to diseases in humans. As one of the key regulatory mechanisms, some TFs control their own expression levels through autogenous regulation. However, identifying autogenous regulation events of TFs has been a tedious task. In this study, we present a straightforward approach that provides a reliable means to identify TF autogenous regulation events. Our method provides a valuable means for understanding the function of this important class of proteins in cells.

**KEYWORDS** transcription factors, autogenous regulation, gene regulation

Transcription is a crucial process for retrieving genomic information, which is largely facilitated by DNA-binding transcription factors (TFs) (1). By controlling specific sets of target genes, TFs regulate diverse biological processes in the cell. However, misregulation of their functions can have far-reaching consequences, including diseases in humans (2). As a result, TFs are tightly controlled within the cell to ensure proper regulation of gene expression.

TFs recognize and bind to short DNA consensus motifs at the promoter of target genes. Subsequently, they recruit chromatin remodelers, chromatin modifiers, mediators, general TFs, and RNA polymerase II to the core promoter to activate (in the case of activators) or repress (in the case of repressors) transcription (3). The transcriptional regulatory activity of TFs can be controlled at both the transcriptional and post-translational levels. At the transcriptional level, many TF genes can be regulated by their own function through autogenous regulation, which is a feedback mechanism that amplifies their transcriptional effect. In addition, TFs' function can also be post-translationally controlled at many levels including nuclear localization (4), ligand binding (5),

Address correspondence to Koon Ho Wong, koonhowong@umac.mo.

Longguang Qin and Shuhui Guo contributed equally to this article. The order of the two co-first authors was based on the amount of intellectual and experimental inputs to the work.

The authors declare no conflict of interest.

See the funding table on p. 9.

dimerization and protein-protein interactions (6), and DNA binding (7). Some of these mechanisms are often coupled with protein modifications such as phosphorylation (8).

Multiple methods are available to study the various regulatory mechanisms of TFs. Western blot analysis can determine protein levels and modifications, while microscopy using fluorescent protein fusions or immunostaining can identify sub-cellular localization. Bimolecular fluorescent complementation is useful for demonstrating interactions between two proteins-of-interest and uncovering their sub-cellular location (9). Chromatin immune-precipitation (ChIP) and ChIP sequencing (ChIP-seq) are widely used to determine DNA binding of TFs (10). However, there is currently no simple way to reveal autogenous regulation events. While the presence of a recognition consensus motif of a TF at its own promoter can suggest autogenous regulation, these motifs are often short and degenerate occurring frequently in the genome. Hence, direct evidence such as promoter truncation analysis with a reporter gene (e.g., GFP or beta-galactosidase), expression analysis of the TF gene in the TF mutant background, or ChIP or ChIP-seq is needed to confirm autogenous regulation. As these experiments are laborious and technically challenging, it is desirable to have reliable methods for predicting autogenous regulation events to better understand TFs.

This study presents a straightforward approach for predicting whether a TF gene is subject to autogenous regulation. The strategy involves creating an experimental strain that expresses an untagged version of the TF gene of interest under a conditional over-expression promoter and an epitope-tagged version of the same TF gene from its own native promoter. The identification of autogenous regulation can be achieved using western blot analysis. We have developed a single-step transformation system and constructed the necessary plasmids to facilitate the generation of experimental strains. The approach has been successfully used for two uncharacterized TFs of secondary metabolism and the widely studied Carbon Catabolite Repressor CreA in the model fungus *Aspergillus nidulans* and is generally applicable to other organisms.

## RESULTS AND DISCUSSION

### Rationale of the strategy

DNA-binding TFs can control their own expression autogenously (Fig. 1A). We devised a simple strategy to identify autogenous regulation simply using western blot analysis. The strategy involves an experimental strain that expresses two versions of the TF of interest: one native form that is expressed from a conditional over-expression promoter and another epitope-tagged version that is expressed from its native wild-type promoter (Fig. 1B). If the TF were autogenously regulated, over-expression of the untagged TF would alter the expression of the epitope-tagged copy, resulting from its transcriptional activity on its own native promoter. Therefore, autogenous regulation of the TF can be evaluated by comparing the levels of the epitope-tagged TF before and after over-expression using western blot analysis (Fig. 1C).

### A collection of plasmids to facilitate the experimental setup of the strategy

We have created a set of universal plasmids that facilitate the generation of a transformation construct (Fig. 2, Step 1) and enable the one-step generation of an experimental strain expressing both tagged and untagged versions of any TF of interest (Fig. 2, Step 2). The plasmids comprise the sequences of a commonly used epitope (e.g., 3xHA, 6xHIS, or 3xFLAG), the *Aspergillus fumigatus pyroA* selectable marker, and an internal region of the *yA* gene for targeting the *yA* genomic locus (11, 12) (Fig. 2, Step 1). The TF gene-of-interest is inserted in between the highly inducible *xylP*(p) promoter (13) and the epitope tag of choice using Isothermal Assembly (14) (or other cloning methods) to yield a transformation construct. When transforming into *nkuAΔ* strains that lack the non-homologous end-joining DNA repair function (15), the constructs can only be integrated via homologous recombination into either the endogenous TF gene or the *yA* gene locus (Fig. 2, Step 2). Integration at *yA* would lead to a strain that

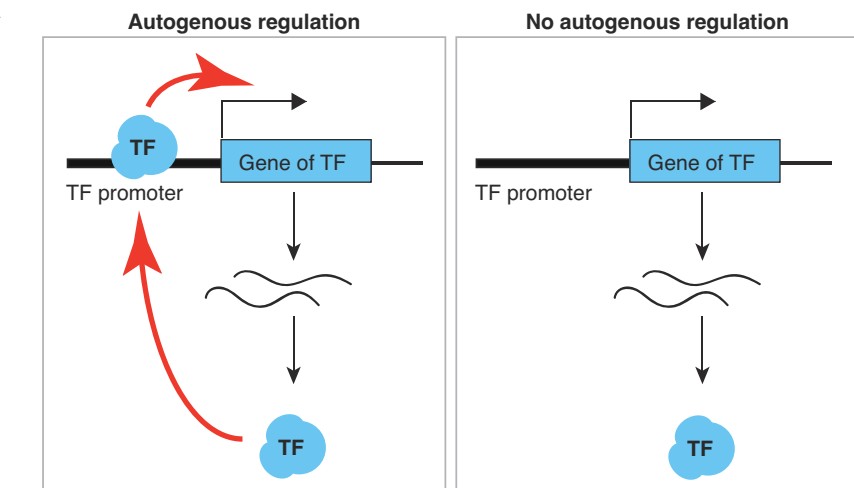

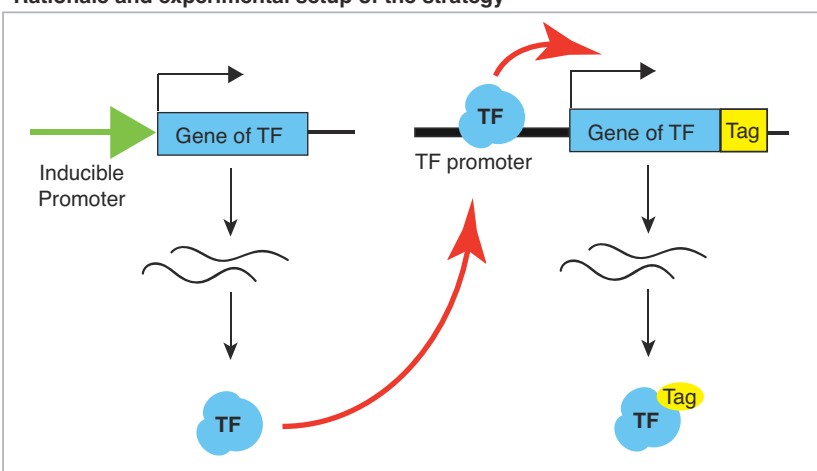

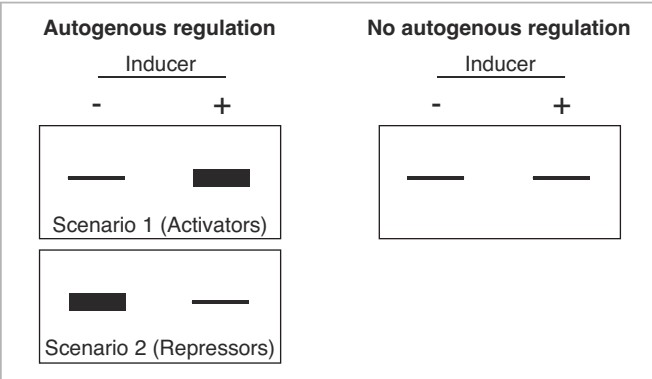

**FIG 1** Rationale of the strategy for identifying autogenous regulation. (A) Schematic diagrams showing a TF gene with and without autogenous regulation. (B) A schematic diagram illustrating the rationale of the strategy and the genetic setup for the experiment strain. (C) Expected Western blot results for a TF with and without autogenous regulation. For TFs with autogenous regulation, one of the two scenarios may be observed with scenarios 1 and 2 indicating positive (e.g., for activators) and negative (e.g., for repressors) autogenous regulation, respectively.

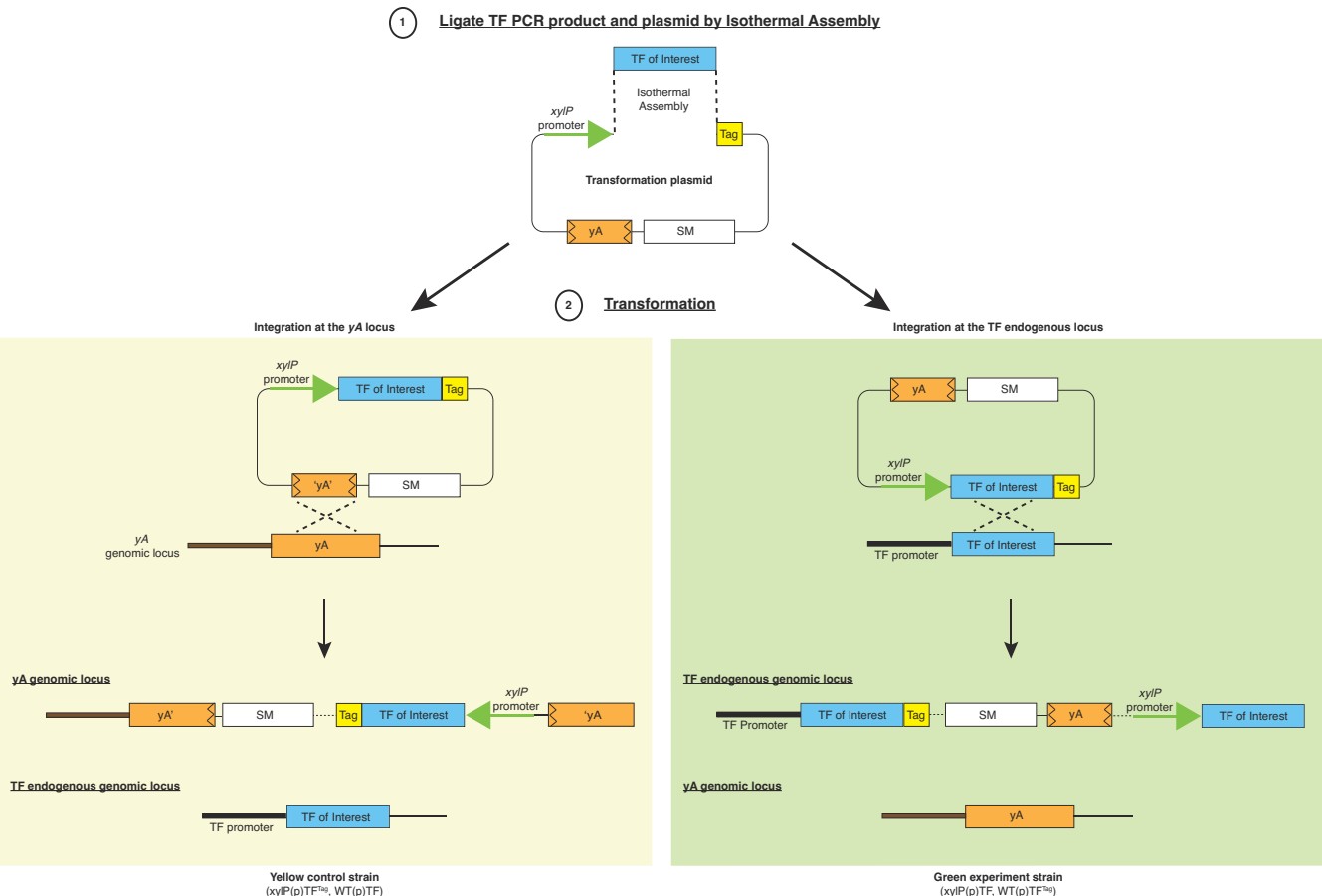

**FIG 2** One-step construction method for the experiment and control strains. A set of plasmids that contains different epitopes (Tag) (e.g., 3xHA, 6xHIS, or 3xFLAG), an internal truncated *yA* gene fragment ("yA"), and a selectable marker (SM) (e.g., *A. fumigatus pyroA*) has been generated to facilitate the creation of an over-expression construct for transformation using Isothermal Assembly or similar cloning methods (Step 1). When transformed into an *nkuAΔ* mutant background, the construct can integrate via homologous recombination at either the *yA* or the TF endogenous genomic locus (Step 2). If the transformation construct is integrated at the *yA* locus, the resulting transformants will yield yellow conidia and express the epitope-tagged TF gene from the inducible *xylP* promoter [*xylP*(p)*TF^Tag*], with the untagged copy being expressed from the native promoter [WT(p)*TF*]. Conversely, if the transformation construct is integrated at the endogenous locus of the TF gene, the transformants will be green, expressing the epitope-tagged TF gene under the control of its native promoter [WT(p)*TF^Tag*], while the untagged copy will be expressed from the *xylP* promoter [*xylP*(p)*TF*].

expresses the epitope-tagged version of the TF-of-interest from the conditional *xylP*(p) over-expression promoter, which is useful for positive control and protein size reference for western blot analysis. On the other hand, if the integration occurs at the endogenous locus, the epitope-tagged TF-of-interest will be expressed from its native promoter, which is the desired experimental strain for investigating autogenous regulation. The site of integration (e.g., endogenous versus *yA*) can be determined from the conidial color of transformants, as integration at the *yA* gene would disrupt its laccase function, which turns yellow conidial pigment into green (16, 17), and confirmed by Southern blot analysis. Consequently, green and yellow transformants are anticipated from the transformation and will be employed as the experimental and control strains, respectively.

## Determining autogenous regulation for two uncharacterized transcription regulators of secondary metabolism gene clusters

We applied the strategy to two uncharacterized *A. nidulans* TFs (AN1678 and AN10295) residing in different secondary metabolite gene clusters, which are transcriptionally silent under most growth conditions (18). Over-expression constructs were created using

the plasmids described above (Fig. 2) for the two TFs and independently transformed into an *nkuAΔ* strain (MH11036—*nkuAΔ*, *pyroA4*, and *riboB2*). Both green (CWF1275 and CWF1276) and yellow transformants (CWF1124 and CWF1277) were obtained, and the integration site in those transformants was confirmed by Southern blot analysis.

Western blot analysis showed that the TFs could be detected in the yellow transformants, which have the epitope-tagged gene fused to the *xylP* promoter, in the presence of the xylose inducer (Fig. 3A and B). In the green experimental strains that express the epitope-tagged TFs from their own native promoter, both TFs were not expressed in the absence of xylose, while distinct protein bands of expected sizes were detected under xylose-inducing conditions (Fig. 3A and B). These results suggest that *AN1678* and *AN10295* are positively regulated by autogenous control. Indeed, ChIP analysis shows strong binding of AN1678[HIS] at its own promoter when expressed (Fig. 3C), validating the western blot result for determining autogenous regulation.

## An example of negative autogenous control by the transcriptional repressor CreA

The strategy was tested on the well-established carbon catabolite repressor, CreA. An experimental strain (i.e., a green transformant carrying the epitope-tagged *creA* gene from the native promoter and untagged *creA* from the *xylP* over-expression promoter) was generated using the method described above. Western blot analysis showed a significant reduction in the level of CreA[HA] when the untagged *creA* gene was over-expressed (Fig. 3D). The reduction is not due to the regulation of *creA* by the addition of xylose (i.e., increased sugar concentration in the culture), as the CreA levels were similar in the control strain [WT(p)*creA*[HA]] that expresses CreA[HA] from its native promoter. Taken together, the western blot results indicate that CreA negatively regulates its own expression. Therefore, the strategy can also identify negative autogenous regulatory activities of TFs.

## Limitations and considerations of the approach

The success of this strategy depends on whether the regulator-of-interest's transcriptional activity can be induced through over-expression. Therefore, it is suitable for TFs whose regulatory function is governed by expression levels and may not work for TFs whose function is controlled post-translationally (e.g., through subcellular localization or post-translational modification) or by specific signals (e.g., a metabolic inducer) not present under the experimental conditions. It is noteworthy that the strong over-expression by the *xylP* promoter and 1% xylose may overwhelm negative regulation imposed on the TF-of-interest, thereby relieving its transcriptional activity to activate its own promoter. Regardless, a negative result from the western blot does not necessarily indicate a lack of autogenous regulation. In such cases, ChIP or ChIP-seq analysis is necessary to determine whether a TF binds to and regulates its own promoter. In addition, the strategy also would not work for TFs whose expression level is below the detection limit of western blot analysis. For these TFs, the strategy can be modified to use more sensitive gene expression detection methods such as reverse transcription PCR (RT-PCR), microscopy (using a fluorescent protein as the tag), or enzymatic assay (using β-glucuronidase or Luciferase as the tag). Lastly, the strategy is also not applicable to fungal species that exhibit the quelling phenomenon (19), which can result in the inactivation of gene expression from transformation constructs.

While the strategy can reveal negative autogenous regulation, it may be constrained by the protein turnover rate of the TF under study. If the protein-of-interest is highly stable, there may be no notable reduction in its level, even if transcriptional repression is induced. Therefore, a slight modification in the experimental setup, such as extending the period after over-expression, is recommended for TFs that are expected to have a negative role.

Our strategy, along with the ChIP/ChIP-seq data, can provide valuable insights into how the TF's function is regulated. For example, a positive western blot result from our

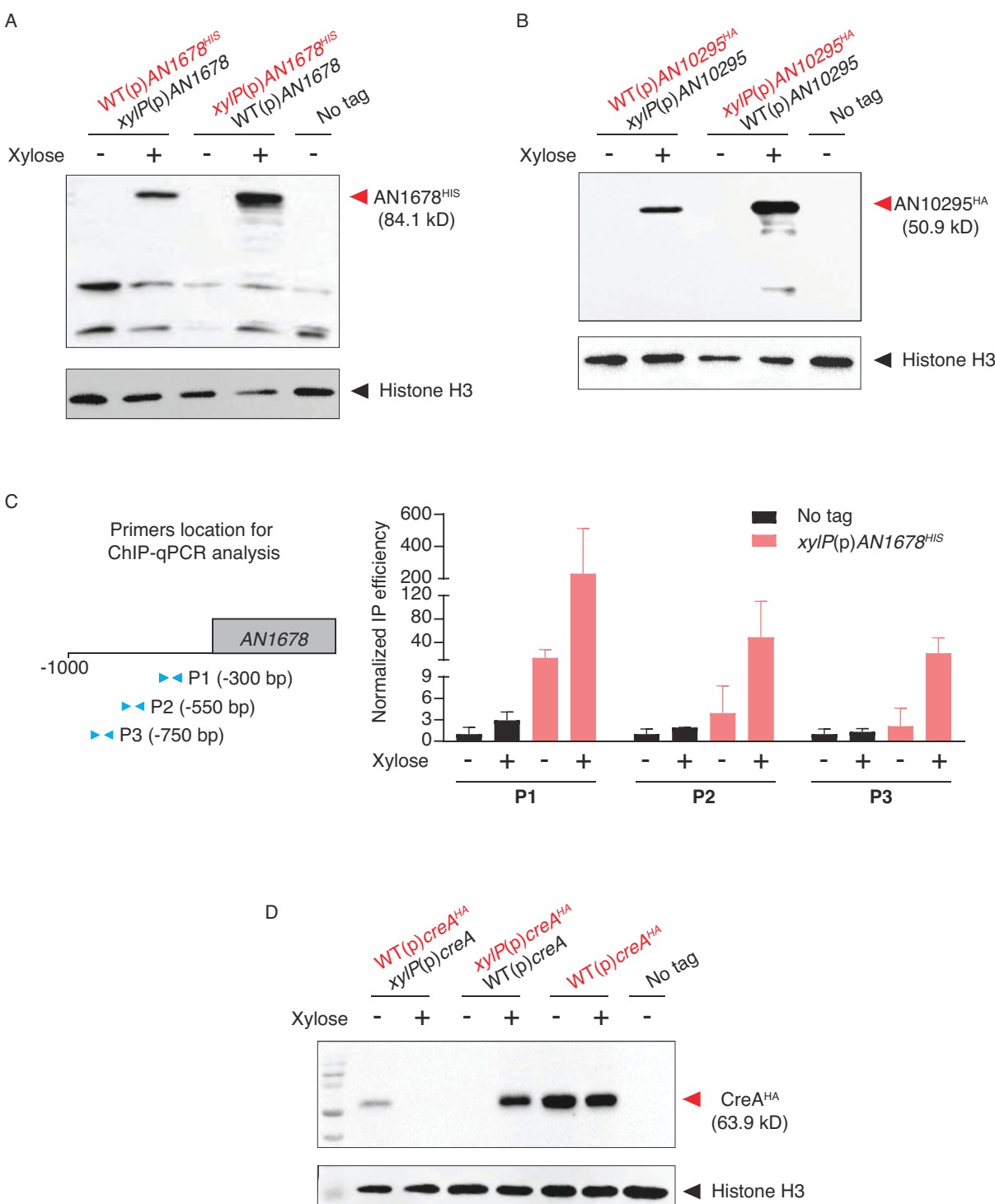

**FIG 3** Implementation of the strategy to study two uncharacterized TFs and the well-established Carbon Catabolite Repressor CreA. Western blot results of the experimental and control (MH11036; No tag) strains for (A) AN1678, (B) AN10295, and (D) CreA with and without xylose induction. (C) ChIP-qPCR analysis shows the binding of AN1678$^{HIS}$ at its own gene promoter. The schematic diagram on the left panel displays the location of primer pairs (blue triangles; P1, P2, and P3) used in the qPCR analysis on DNA immune precipitated from the chromatin of wild type (No tag; black bars) (MH11036) and the *xylP*(p)*AN1678*$^{HIS}$ strain (red bars) (CWF753) using the anti-HIS antibody shown on the right. The approximate locations of the primer pairs are shown in parentheses. (D) The WT(p)*creA*$^{HA}$ strain that expressed *creA*$^{HA}$ from its native promoter was included as a positive control. The relevant genotype of the strains with respect to the TF gene of interest is shown with the epitope-tagged version highlighted in red, while their full genotypes can be found in Table 2. Red and black arrows in (A), (B), and (D) indicate the expected locations of the protein-of-interest and histone H3 based on their molecular weight. The expected protein sizes are given in parentheses.

strategy may possibly be due to an indirect effect of a regulatory loop involving two or more TFs. In this case, a negative ChIP/ChIP-seq experiment would be expected. In addition, if a negative western blot result is accompanied by positive ChIP/ChIP-seq evidence for binding at the TF's own promoter, it suggests that the TF's regulation operates at the post-translational modification level rather than through expression, nuclear localization, or DNA binding. Conversely, if both western blot and ChIP/ChIP-seq results are positive, it indicates that the TF's regulatory function is, at least in part, influenced by its expression.

## MATERIALS AND METHODS

### Plasmid construction

Plasmids for constructing a strain to over-express an epitope-tagged TF-of-interest were created by first inserting the epitope (3xHA, 6xHIS, or 3xFLAG) and the terminator sequence of *Saccharomyces cerevisiae ADH1* [*ADH1*(T)] after the promoter of the *xylP* gene of *Penicillium chrysogenum* in the construct XPyA-3 (CWB13) that also carries a 724 bp fragment of the internal coding sequence of *yA* for targeting and the *pyroA* gene of *A. fumigatus* for selection. The epitope tag and *ADH1*(T) DNA fragment were amplified by PCR using pFA6a-3xHA-Bar, pFA6a-6Gly-3xFLAG-Bar, and pFA6a-6×Gly-His-Bar as the DNA template (20). The PCR product was inserted into the XPyA-3 using Isothermal Assembly (14). The resultant vector (XPyA-3–1) was linearized and fused to the PCR product of the TF-of-interest coding region using Isothermal Assembly (Fig. 2) as described previously (20). The plasmids for generating over-expression epitope-tagging strains are freely available upon request (Table 1).

### Strain construction

*A. nidulans* transformation was performed as described previously (21). The over-expression constructs were transformed into an *nkuΔ* strain (MH11036—*nkuAΔ*, *riboB2*, *pyroA4*) (15). Subsequently, green and yellow transformants were selected and purified using pyridoxine-deficient media, and the integration site (e.g. *yA* or the native locus) was verified through Southern blot analysis. Table 2 summarizes the strains that were produced and utilized in this study.

### Growth conditions

Experimental strains were grown in 100 mL of *A. nidulans* nitrogen-free minimal media (ANM) (23) with 1% glucose and 10 mM ammonium tartrate as the sole carbon and nitrogen source, respectively, at 37°C in a shaking incubator (220 rpm) for 16 hours. To induce the over-expression of the TF, xylose was added to the culture at a final concentration of 1% for 4 hours, while glucose was added to the non-induced control. The mycelia were then harvested, washed with ice-cold water, pressed-dried, and subsequently stored at −80°C until protein extraction.

### Protein extraction and western blot analysis

Total proteins were extracted as described previously (11). Fifty micrograms of total protein was used for western blot analysis using 1:3,000 of anti-HA antibody for (sc7392,

**TABLE 1** Plasmids for generating the experimental strains for determining autogenous regulation

| Plasmid name | Description |
|---|---|
| CWB87 (pXPyA-HA) | For tagging with 3xHA. Contains the *pyroA* selectable marker and an internal fragment of the *yA* gene for targeting to the *yA* locus. |
| CWB871 (pXPyA-FLAG) | For tagging with 3xFLAG. Contains the *pyroA* selectable marker and an internal fragment of the *yA* gene for targeting to the *yA* locus. |
| CWB872 (pXPyA-HIS) | For tagging with 6xHIS. Contains the *pyroA* selectable marker and an internal fragment of the *yA* gene for targeting to the *yA* locus. |

**TABLE 2** Strains used in this study

| Strain name | Genotype | Remark | Reference |
|---|---|---|---|
| MH11036 | $veA_1$; pyroA4; riboB2; nkuA::argB | Parental control strain | (15) |
| CWF1124 | yA::xylP(p)AN1678-HIS- AfpyroA; pyroA4; riboB2; nkuA::argB | Yellow transformant with the construct integrated at the yA locus. | This study |
| CWF1275 | AN1678::xylP(p)AN1678-HIS-AfpyroA; pyroA4; riboB2; nkuA::argB | Green transformant with the construct integrated at the AN1678 locus. | This study |
| CWF753 | veA1; yA::xylP(p)AN1678-HIS- AfpyroA; pyroA4; riboB2; nkuA::argB | Yellow transformant with the construct integrated at the yA locus. | This study |
| CWF1276 | AN10295::xylP(p)AN10295-3xHA-AfpyroA; pyroA4; riboB2; nkuA::argB | Green transformant with the construct integrated at the AN10295 locus. | This study |
| CWF1277 | yA::xylP(p)AN10295-3xHA-AfpyroA; pyroA4; riboB2; nkuA::argB | Yellow transformant with the construct integrated at the yA locus. | This study |
| CWF1278 | yA::xylP(p)creA-3xHA-AfpyroA; pyroA4; riboB2; nkuA::argB | Yellow transformant with the construct integrated at the yA locus. | This study |
| CWF1279 | creA::xylP(p)creA-3xHA-AfpyroA; pyroA4; riboB2; nkuA::argB | Green transformant with the construct integrated at the creA locus. | This study |
| CWF191 | creA-3xHA-Bar; pyroA4; riboB2; nkuA::argB; veA1 | Positive control strain with $creA^{HA}$ expressed from the creA promoter. | (22) |

Santa Cruz) or 1:10,000 for anti-H3 antibody (ab1791, Abcam) as the primary antibody, and 1:5,000 horseradish peroxidase-conjugated anti-mouse (AP124P, Sigma-Aldrich) or anti-rabbit antibody (AP132P, Sigma-Aldrich) as the secondary antibody. Chemiluminescence detection was performed using the Clarity ECL substrate kit (BIO-RAD, USA).

## Conclusions

This work describes a straightforward strategy for detecting autogenous regulation of TFs and provides a set of plasmids that simplify the construction of experimental strains in fungi. The strategy is versatile, enabling the detection of both positive and negative autogenous regulatory activities, and the underlying principle is generally applicable to other organisms for identifying autogenous regulation activities of TFs.

## ACKNOWLEDGMENTS

We thank members of the Wong Laboratory for their helpful comments on the work.

This work was supported by the Science and Technology Development Fund of Macau S.A.R. (FDCT 0033/2021/A1 and FDCT 0099/2022/A2), the Research Services and Knowledge Transfer Office of the University of Macau (MYRG2022-00107-FHS), and Dr. Stanley Ho Medical Development Foundation (SHMDF-OIRFS/2022/001).

## AUTHOR AFFILIATIONS

[1]Faculty of Health Sciences, University of Macau, Avenida da Universidade, Taipa, Macau, China
[2]Gene Expression, Genomics and Bioinformatics core, Faculty of Health Sciences, University of Macau, Avenida da Universidade, Taipa, Macau, China
[3]Institute of Translational Medicine, Faculty of Health Sciences, University of Macau, Avenida da Universidade, Taipa, Macau, China
[4]MoE Frontiers Science Center for Precision Oncology, University of Macau, Taipa, Macau, China

## AUTHOR ORCIDs

Koon Ho Wong  http://orcid.org/0000-0002-9264-5118

## FUNDING

| Funder | Grant(s) | Author(s) |
|---|---|---|
| Universidade de Macau (UM) | MYRG2022-00107-FHS | Koon Ho Wong |
| Fundo para o Desenvolvimento das Ciências e da Tecnologia (FDCT) | FDCT 0033/2021/A1 | Koon Ho Wong |
| Dr. Stanley Ho Medical Development Foundation | SHMDF-OIRFS/2022/001 | Koon Ho Wong |

## AUTHOR CONTRIBUTIONS

Longguang Qin, Conceptualization, Formal analysis, Investigation, Methodology | Shuhui Guo, Data curation, Investigation, Resources | Ang Li, Formal analysis, Investigation | Lu Fan, Investigation | Kaeling Tan, Resources, Supervision | Koon Ho Wong, Conceptualization, Formal analysis, Funding acquisition, Investigation, Methodology, Project administration, Resources, Supervision, Writing – original draft, Writing – review and editing

## DATA AVAILABILITY

The plasmids for generating experimental strains (Table 1) will be freely available upon request or can be obtained from the Fungal Genetics Stock Center.

## ADDITIONAL FILES

The following material is available online.

Open Peer Review

**PEER REVIEW HISTORY (review-history.pdf).** An accounting of the reviewer comments and feedback.

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
