## [Reviewer comments · Microbiology Spectrum]

Microbiology Spectrum

An effective strategy for identifying autogenous regulation of TFs in filamentous fungi

Longguang Qin, Shuhui Guo, Ang Li, Lu Fan, Kaeling Tan, and Koon Ho Wong

Corresponding Author(s): Koon Ho Wong, University of Macau

Review Timeline:

Submission Date:	June 5, 2023
Editorial Decision:	July 9, 2023
Revision Received:	August 24, 2023
Accepted:	September 8, 2023

Editor: Gustavo Goldman

Reviewer(s): Disclosure of reviewer identity is with reference to reviewer comments included in decision letter(s). The following individuals involved in review of your submission have agreed to reveal their identity: Luis F Larrondo (Reviewer #2)

Transaction Report:

DOI: <https://doi.org/10.1128/spectrum.02347-23>

July 9, 2023

Prof. Koon Ho Wong
University of Macau
Taipa
Macau

Re: Spectrum02347-23 (**An effective strategy for identifying autogenous regulation of TFs in filamentous fungi**)

Dear Prof. Koon Ho Wong:

Your manuscript has been reviewed by three reviewers who suggested many modifications. Please, submit a revised version together with a rebuttal letter addressing point-by-point raised by each reviewer.

Link Not Available

Sincerely,

Gustavo Goldman

Journals Department
Reviewer comments:

Reviewer #1 (Comments for the Author):

The manuscript by Qin et al. describes a simple but elegant method to assess auto-regulation by transcription factors. The method is highly applicable to many organisms with minor modifications to selectable markers and targeting strategy. The manuscript is well written and clear. A few comments are provided below for the authors' consideration.

1. On lines 117-119, the promoter/transcription factor/tag combinations are mixed up. Should it be that tagged version is expressed from the native promoter and the untagged version from the inducible xylP promoter?
2. Throughout the manuscript the word blot is variably capitalised. It should be 'blot'.

3. In the Limitations section it would be prudent to make it clear that where auto-regulation is observed, it may not be direct. Regulatory loops involving two or more transcription factors may produce a similar result.
4. The manuscript mentions a number of different possible epitope tags but it is not clear if they have all been tested. Are there advantages to one over the others? Only constructs involving HA have been shown in this manuscript.
5. On line 218, the word 'southern' should be capitalised.
6. Table 1 does not list any of the creA strains used in this study.
7. The sentence beginning 'When transform...' on line 318 needs editing.
8. There is no description of the strain in Fig 3C with the genotype WT)pcreA-HA. Moreover, the figure legend requires editing to fully describe the figure, the various panels, strains and controls.
9. Do the authors have a feel for how sensitive this assay may be? Might a PCR-based method looking at transcription be more sensitive and quantitative?

Reviewer #2 (Comments for the Author):

The work by Qin and cols, describe a simple -but useful- experimental approach to inquire about autoregulation at the transcriptional level, of transcription factors of interest. The approach is compatible with visualizing autogenous regulation that could correspond to either positive or negative auto feedback loops.

There are several aspects of the MS that could be improved for clarity.

1- It is surprising that no expression is seen for the tagged TFs (expressed from by their own promoters), in the absence of the inducer xylose (i.e., Fig 3A and 3B, first lane in each case). Is this because chemiluminescence was set at a low level (not to overexpose the xylose-induced tagged TF (lanes 4 in both figs)?

A classic practice is to tag a TF of interest at its endogenous locus: those endogenous levels of the tagged version are normally enough not only to be detected by Western blot, but also to do IPs, CHIPseq etc. Therefore, it is rather peculiar that the authors are not seeing any band for both of the TF under analysis. Does this relate to the type of TF under study (secondary metabolite cluster)? From different transcriptomic studies the authors ought to know conditions where the FPKM suggest normal expression levels. Running western blots under those conditions, to show that the autogenous regulation can take bring from normal to high (and not just from low to normal) should be trivial.

2- Related to the prior point: although Fig 3C shows that the strategy can detect a case of an auto negative feedback loop, what happens in cases where the expression of the TF of interest is too low (as in Fig 3A and 3B)? Will that "baseline" expression jeopardize detecting the negative effect of the TF on its own expression?

3- One of the messages of the work is that with only one plasmid it is possible to generate two types of strains: one with a tagged inducible TF, and another one (experimentally valuable) where the tagged version at the endogenous locus will allow exploring the effect of overexpressing the same TF from a different promoter.

Yet, an alternative strategy (involving one type of strain and two constructs), implies changing the promoter of the TF of interest (by an inducible one), and integrating a reporter gene (GUS, Luciferase, or destabilized GFP) under the control of the TF's promoter. Importantly, such a strategy is less cumbersome when analyzing results (measuring fluorescence/ luminescence versus western blots), and can be adopted to track changes of interest in the transcription of the TF of interest in a continuous fashion, and under multiple experimental conditions. Could the authors comment on the pros and cons of each strategy?

4- Related to the previous point, could the authors comment on whether utilizing GFP/RFP as a tag would also be a valid option (in that case if would be possible to easily measure + or - autofeedback loops) with only one plasmid construct.

5- Line 71 "In addition, TFs' function can also be controlled through nuclear localization (4), which is often coupled with protein modifications such as phosphorylation (5)."

While the statement is correct, this part of the introduction falls short in describing the plethora of mechanisms affecting TF activity. Please improve this in 1-2 sentences, including additional references.

6- Line 80 : "Hence, direct evidence such as promoter truncation analysis with a reporter gene (e.g. GFP or beta galactosidase), expression analysis of the TF gene in the TF mutant background, or chromatin immuno-precipitation (ChIP) or ChIP-sequencing (ChIP-seq) to confirm autogenous regulation."

This phrase is missing the verb/action (i.e. ...direct evidence..... is needed"

7- Line 117: ". The tagged version is expressed from the highly inducible xylP(p) promoter (Zadra et al., 2000), while the untagged version is expressed from the TF's native promoter"

This phrase is confusing as it appears to indicate that the plasmids are design to yield only this type of strain. Yet, while reading more into the text, the different integrations and results are explained.

8- Line 150: "On the other hand, both TFs were not expressed without xylose induction in the green transformants (i.e. the experimental strains that express the epitope-tagged TFs from their own native promoter). However, under xylose inducing conditions, a distinct protein band was detected for both TFs that were of the same size as the positive controls (i.e. yellow transformants) (Fig. 3A and B)."

This paragraph reads more complicated that it should. Please consider rewriting and eliminating the "however" in the last phrase.

9- Line 65: "(for activators) or repress (for repressors) transcription"

Change "for " for "in the case of"

Reviewer #3 (Comments for the Author):

The ms by Qin et al describes a relatively simple method to study the existence of autogenous regulation of transcription factors. The idea sounds good and straightforward. The presentation of the results is somewhat strange to me, as I would present first the control example (in this case, CreA) to demonstrate the functionality of the system, and then the unknown test examples. The xylP promoter is titratable. If the systems works well enough, authors should be able to see an effect of using different concentrations of xylose.

What I do miss in this ms is some further support of the data presented, i.e. either using one additional characterized TF as a control (this could positive regulation), or some ChIP data on the other two uncharacterized genes to confirm the data obtain with this system.

Maybe authors can include a comment about one additional limitation in some fungal species: quelling (Romano N, Macino G. Quelling: transient inactivation of gene expression in *Neurospora crassa* by transformation with homologous sequences. *Mol Microbiol.* 1992 Nov;6(22):3343-53. doi: 10.1111/j.1365-2958.1992.tb02202.x. PMID: 1484489.)

Minor comments:

Table 1. genotype description is somewhat confussing, as according to the genotypes reported the genes tagged are in all strains under the control of the xylP(p). Additionally, creA strains used in this study are missing from this table.

L222-224. I guess that authors used glucose to grow the strains during the first 16 h.

Staff Comments:

Preparing Revision Guidelines

Please return the manuscript within 60 days; if you cannot complete the modification within this time period, please contact me. If you do not wish to modify the manuscript and prefer to submit it to another journal, please notify me of your decision immediately so that the manuscript may be formally withdrawn from consideration by Microbiology Spectrum.

Point-by-point response to reviewers' comments

Reviewer #1 (Comments for the Author):

The manuscript by Qin et al. describes a simple but elegant method to assess auto-regulation by transcription factors. The method is highly applicable to many organisms with minor modifications to selectable markers and targeting strategy. The manuscript is well written and clear. A few comments are provided below for the authors' consideration.

Response

We thank this reviewer for his/her time and the helpful comments. We have carefully addressed all his/her comments as described below.

1. On lines 117-119, the promoter/transcription factor/tag combinations are mixed up. Should it be that tagged version is expressed from the native promoter and the untagged version from the inducible xyIP promoter?

Response

We are sorry for the confusion. This sentence should be describing the transformation construct. We have deleted the sentence and re-written the paragraph to clarify the information.

2. Throughout the manuscript the word blot is variably capitalised. It should be 'blot'.

Response

Corrected.

3. In the Limitations section it would be prudent to make it clear that where auto-regulation is observed, it may not be direct. Regulatory loops involving two or more transcription factors may produce a similar result.

Response

The following statements has been added to suggest this possibility in the Limitations section on lines 203-207: *“Our strategy, along with the ChIP/ChIPseq data, can provide valuable insights into how the TF's function is regulated. For example, a positive Western blot result from our strategy may possibly be due to an indirect effect of a regulatory loop involving two or more transcription factors. In this case, a negative ChIP/ChIPseq experiment would be expected.”*

4. The manuscript mentions a number of different possible epitope tags but it is not clear if they have all been tested. Are there advantages to one over the others? Only constructs involving HA have been shown in this manuscript.

Response

The selected epitopes are widely used in biochemical experiments, and there are many high-quality commercial antibodies available for them, making it easy and convenient for users to employ our strategy in their research. To increase the flexibility and robustness of our strategy and account for the possibility of the introduced foreign peptide affecting the 3-dimensional shape and function of TFs, we developed plasmids for the three different epitopes. The detection sensitivities for the three epitopes may vary depending on the antibody used. It is not possible to predict which is superior for a given TF without empirical testing and the result is expected to be TF-specific. However, epitope tagging is a standard practice in biochemical experiments, and we have successfully utilized the 3xHA, 6xHIS, and 3xFLAG epitopes in Western blot and ChIP/ChIP-seq experiments in our laboratory. Examples of the results for 3xHA and 6xHIS are presented in Figure 3.

The sentence on lines 120-123 has been revised as follows “*The plasmids comprise the sequences of a commonly used epitope (e.g. 3xHA, 6xHIS or 3xFLAG), the *Aspergillus fumigatus* *pyroA* selectable marker and an internal region of the *yA* gene for targeting to the *yA* genomic locus (11, 12)(Fig. 2 Step 1).*”

5. On line 218, the word 'southern' should be capitalised.

Response

Corrected.

6. Table 1 does not list any of the *creA* strains used in this study.

Response

The genotype information for the *creA* strains have been added to Table 1.

7. The sentence beginning 'When transform...' on line 318 needs editing.

Response

Edited.

8. There is no description of the strain in Fig 3C with the genotype WT(p)creA-HA. Moreover, the figure legend requires editing to fully describe the figure, the various panels, strains and controls.

Response

The description for the WT(p)creA^{HA} strain has been included, and the legend has been edited accordingly. The revised legend on lines 370-385 now reads as follows: “*Western blot results of the experimental and control (MHI1036; No tag) strains for (A) AN1678, (B) AN10295 and (D) CreA with and without xylose induction. (C) ChIP-qPCR analysis showing AN1678^{HIS} binding to its own gene promoter. The schematic diagram on the left panel displays the location of primer pairs (blue triangles; P1, P2 and P3) used in the qPCR analysis on DNA immune-precipitated from the chromatin of wildtype (No tag; black bars) and the xylP(p)AN1678^{HIS} strain (red bars) using the anti-HIS antibody shown on the right. The approximate locations of the primer pairs are shown in parentheses. (D) The WT(p)creA^{HA} strain that expressed creA^{HA} from its native promoter was included as a positive control. The relevant genotype of the strains with respect to the TF gene of interest is shown with the epitope-tagged version highlighted in red, while their full genotypes can be found in Table 1. Red and black arrows in (A), (B) and (D) indicate the expected locations of the protein-of-interest and histone H3 based on their molecular weight. The expected protein sizes are given in parentheses.*”

9. Do the authors have a feel for how sensitive this assay may be? Might a PCR-based method looking at transcription be more sensitive and quantitative?

Response

The sensitivity of the Western blot assay is largely dependent on the antibody used, which is why we selected commonly used epitopes that have numerous high-quality commercial antibodies available. Although a PCR-based method is more sensitive and quantitative than Western blot analysis, it involves more cumbersome and technically demanding steps such as mRNA extraction and cDNA synthesis, and can also be more expensive. Additionally, the PCR-based method requires a pair of primers that can differentiate between the transcripts expressed from the native wildtype promoter and the inducible promoter, which can be challenging to design and, in some cases, not possible.

This suggestion has been included in the revised manuscript on lines 189-193 as follows: “*In addition, the strategy also would not work for TFs whose expression level is below the detection limit of Western blot analysis. For these TFs, the strategy can be modified to use a more sensitive protein detection methods such as RT-PCR, microscopy (using a fluorescent protein as the tag) or enzymatic assay (using β -glucuronidase or Luciferase as the tag).*”

Reviewer #2 (Comments for the Author):

The work by Qin and cols, describe a simple -but useful- experimental approach to inquire about autoregulation at the transcriptional level, of transcription factors of interest. The approach is compatible with visualizing autogenous regulation that could correspond to either positive or negative auto feedback loops. There are several aspects of the MS that could be improved for clarity.

Response

We thank this reviewer for his/her time and the constructive comments. We have carefully addressed all his/her comments as described below.

1- It is surprising that no expression is seen for the tagged TFs (expressed from by their own promoters), in the absence of the inducer xylose (i.e., Fig 3A and 3B, first lane in each case). Is this because chemiluminescence was set at a low level (not to overexpose the xylose-induced tagged TF (lanes 4 in both figs)?)

A classic practice is to tag a TF of interest at its endogenous locus: those endogenous levels of the tagged version are normally enough not only to be detected by Western blot, but also to do IPs, CHIPseq etc. Therefore, it is rather peculiar that the authors are not seeing any band for both of the TF under analysis. Does this relate to the type of TF under study (secondary metabolite cluster)? From different transcriptomic studies the authors ought to know conditions where the FPKM suggest normal expression levels. Running western blots under those conditions, to show that the autogenous regulation can take bring from normal to high (and not just from low to normal) should be trivial.

Response

The reviewer is correct that the no expression is related to the regulation of genes within secondary metabolism clusters, which are silent. Therefore, it is not unexpected that the two secondary metabolism TFs could not be detected on Western blot.

We have revised the following sentence and added a reference about secondary metabolism regulation to clarify the no expression observation on lines 144-146 as follows: "*We applied the strategy to two uncharacterized A. nidulans TFs (AN1678 and AN10295) residing in different secondary metabolite gene clusters, which are transcriptionally silent under most growth conditions (18).*"

2- Related to the prior point: although Fig 3C shows that the strategy can detect a case of an auto negative feedback loop, what happens in cases where the expression of the TF of interest is too low (as in Fig 3A and 3B)? Will that "baseline" expression jeopardize detecting the negative effect of the TF on its own expression?

Response

Because our strategy relies on changes in protein levels as a proxy for autogenous regulation, it may not be applicable for TFs whose expression cannot be detected (i.e. too low) using this

method. However, negative-acting transcription factors are often expressed at high levels to exert repressive functions, so most repressors should not be affected by this limitation. Nevertheless, it is important to acknowledge this possibility, and we have included a statement about this in the limitation section on lines 189-193 as follows: “*In addition, the strategy also would not work for TFs whose expression level is below the detection limit of Western blot analysis. For these TFs, the strategy can be modified to use a more sensitive protein detection methods such as RT-PCR, microscopy (using a fluorescent protein as the tag) or enzymatic assay (using β -glucuronidase or Luciferase as the tag).*”.

3- One of the messages of the work is that with only one plasmid it is possible to generate two types of strains: one with a tagged inducible TF, and another one (experimentally valuable) where the tagged version at the endogenous locus will allow exploring the effect of overexpressing the same TF from a different promoter.

Yet, an alternative strategy (involving one type of strain and two constructs), implies changing the promoter of the TF of interest (by an inducible one), and integrating a reporter gene (GUS, Luciferase, or destabilized GFP) under the control of the TF's promoter. Importantly, such a strategy is less cumbersome when analyzing results (measuring fluorescence/ luminescence versus western blots), and can be adopted to track changes of interest in the transcription of the TF of interest in a continuous fashion, and under multiple experimental conditions. Could the authors comment on the pros and cons of each strategy?

Response

We agree with this reviewer that the suggested strategy is another feasible approach for identifying autogenous regulation events, although we respectfully hold a different opinion about fluorescence and luminescence assays being less cumbersome than Western blot analysis. We believe that the relative ease of these experiments may vary depending on the laboratory. For instance, fluorescence assays require a fluorescent microscope that may not be available in all laboratories. Regardless, the reviewer is correct that the suggested strategy provides the ability to monitor changes continuously and under multiple experimental conditions. However, constructing an experimental strain for this strategy is relatively slow and laborious, involving two plasmid constructions and two separate, stepwise transformations that require different protoplasts. In contrast, our strategy only requires a single plasmid construct for each TF of interest, and the same wildtype protoplast preparation can be used to generate experimental strains with different plasmid constructs for different TFs. Consequently, our strategy is suitable for high-throughput screening of autogenously regulated TFs, which is not practical with the suggested approach. Therefore, we think that both strategies are valid ways for identifying autogenous regulation events, each with their own merits.

4- Related to the previous point, could the authors comment on whether utilizing GFP/RFP as a tag would also be a valid option (in that case if would be possible to easily measure + or - autofeedback loops) with only one plasmid construct.

Response

Yes, the method can also use GFP/RFP as the tag for detection by Western blot analysis using commercially available antibodies against GFP/RFP. If GFP/RFP tag is used, autogenous regulation can be detected by Western blot or fluorescence microscopy. A discussion has been added to include this helpful suggestion on lines 189-193 as follows: *“In addition, the strategy also would not work for TFs whose expression level is below the detection limit of Western blot analysis. For these TFs, the strategy can be modified to use a more sensitive protein detection methods such as RT-PCR, microscopy (using a fluorescent protein as the tag) or enzymatic assay (using β -glucuronidase or Luciferase as the tag).”*

5- Line 71 "In addition, TFs' function can also be controlled through nuclear localization (4), which is often coupled with protein modifications such as phosphorylation (5)."

While the statement is correct, this part of the introduction falls short in describing the plethora of mechanisms affecting TF activity. Please improve this in 1-2 sentences, including additional references.

Response

The statement is now revised as the following (lines 69-73): *“In addition, TFs' function can also be post-translationally controlled at many levels including nuclear localization (4), ligand binding (5), dimerization and protein-protein interactions (6), and DNA-binding (7). Some of these mechanisms are often coupled with protein modifications such as phosphorylation (8).”*

6- Line 80 : "Hence, direct evidence such as promoter truncation analysis with a reporter gene (e.g. GFP or beta galactosidase), expression analysis of the TF gene in the TF mutant background, or chromatin immuno-precipitation (ChIP) or ChIP-sequencing (ChIP-seq) to confirm autogenous regulation."

This phrase is missing the verb/action (i.e. ...direct evidence..... is needed"

Response

Corrected.

7- Line 117: ". The tagged version is expressed from the highly inducible xylP(p) promoter (Zadra et al., 2000), while the untagged version is expressed from the TF's native promoter" This phrase is confusing as it appears to indicate that the plasmids are design to yield only this type of strain. Yet, while reading more into the text, the different integrations and results are explained.

Response

We are sorry for the confusion. This reviewer is right that the sentence is supposed to describe the plasmid construct. We have now deleted the sentence and re-wrote the paragraph to clarify the information.

8- Line 150: "On the other hand, both TFs were not expressed without xylose induction in the green transformants (i.e. the experimental strains that express the epitope-tagged TFs from their own native promoter). However, under xylose inducing conditions, a distinct protein band was detected for both TFs that were of the same size as the positive controls (i.e. yellow transformants) (Fig. 3A and B)."

This paragraph reads more complicated than it should. Please consider rewriting and eliminating the "however" in the last phrase.

Response

The paragraph (lines 154-157) is revised and now reads as *“In the green experimental strains that express the epitope-tagged TFs from their own native promoter, both TFs were not expressed in the absence of xylose, while distinct protein bands of expected sizes were detected under xylose inducing conditions (Fig. 3A and B).”*.

9- Line 65: "(for activators) or repress (for repressors) transcription"
Change "for " for "in the case of"

Response

Changed.

Reviewer #3 (Comments for the Author):

The ms by Qin et al describes a relatively simple method to study the existence of autogenous regulation of transcription factors.

The idea sounds good and straightforward. The presentation of the results is somewhat strange to me, as I would present first the control example (in this case, CreA) to demonstrate the functionality of the system, and then the unknown test examples.

Response

We thank this reviewer for his/her time and the constructive comments.

The suggested presentation (first begin with CreA) is indeed an alternative flow we considered initially. However, we feel that it would be helpful to readers to recognize that the strategy can be used for both positive- and negative-acting transcription factors. Therefore, we decided to introduce our results in that order. Nevertheless, this reviewer is correct that it is important to first establish that the strategy is functioning. To address this, we have now added ChIP evidence (lines 159-161, Fig. 3C), as suggested by this reviewer, to demonstrate that the method indeed can detect autogenous regulation events. With this new direct experimental support, we would prefer to keep the current presentation order if this reviewer does not object.

The xylP promoter is titratable. If the system works well enough, authors should be able to see an effect of using different concentrations of xylose.

Response

This reviewer's expectation is reasonable. As our aim is to have a simple method, we decided to use only one xylose concentration and want to pick a concentration likely to work for most TFs. We chose 1% xylose chosen as it can induce strong TF expression. This high TF induction has an advantage in which it may overwhelm any negative regulation (e.g. phosphorylation, interaction with a negative regulator or sequestration at the cytoplasm, etc.) imposed on the TF-of-interest through a titration effect. Consequently, even for TFs whose transcriptional activity is usually prevented under the experimental condition used, their activity could be relieved in our system, activating its own promoter if they are subjected to autogenous regulation.

A statement has been added to the Limitations and Considerations section on lines 183-186 to explain the choice of 1% xylose for induction as follows: "*It is noteworthy that the strong over-expression by the xylP promoter and 1% xylose may overwhelm negative regulation imposed on the TF-of-interest, thereby relieving its transcriptional activity to activate its own promoter.*"

What I do miss in this ms is some further support of the data presented, i.e. either using one additional characterized TF as a control (this could positive regulation), or some ChIP data on the other two uncharacterized genes to confirm the data obtain with this system.

Response

We have now added ChIP evidence to demonstrate that AN1678 strongly binds to its own promoter, validating that the method can indeed identify autogenous regulation events. The result is presented in Figure 3C and described on lines 159-161 as follows: "*Indeed, ChIP analysis shows strong binding of AN1678^{HIS} at its own promoter when expressed (Fig. 3C), validating the Western blot result for determining autogenous regulation.*"

Maybe authors can include a comment about one additional limitation in some fungal species: quelling (Romano N, Macino G. Quelling: transient inactivation of gene expression in *Neurospora crassa* by transformation with homologous sequences. *Mol Microbiol.* 1992 Nov;6(22):3343-53. doi: 10.1111/j.1365-2958.1992.tb02202.x. PMID: 1484489.)

Response

The following statement has been added to the Limitations section on lines 193-196: "*Lastly, the strategy is also not applicable to fungal species that exhibit the quelling phenomenon (19), which can result in the inactivation of gene expression from transformation constructs.*"

Minor comments:

Table 1. genotype description is somewhat confusing, as according to the genotypes reported the genes tagged are in all strains under the control of the xylP(p). Additionally, *creA* strains used in this study are missing from this table.

Response

We are sorry for the confusion. Details with respect to where the transformation construct is integrated in the strains have been added to the table. The genotype information for the *creA* strains is also added.

L222-224. I guess that authors used glucose to grow the strains during the first 16 h.

Response

Yes, this is correct. The information has been added to lines 239-242.

September 8, 2023

Prof. Koon Ho Wong
University of Macau
Taipa
Macau

Re: Spectrum02347-23R1 (**An effective strategy for identifying autogenous regulation of TFs in filamentous fungi**)

Dear Prof. Koon Ho Wong:

Your manuscript is now ready for publication. Congratulations !!!

Your manuscript has been accepted, and I am forwarding it to the ASM Journals Department for publication. You will be notified when your proofs are ready to be viewed.

Sincerely,

Gustavo Goldman
Editor, Microbiology Spectrum
